# Variational PDEs for Acceleration on Manifolds and Application to Diffeomorphisms

**Ganesh Sundaramoorthi**
United Technologies Research Center
East Hartford, CT 06118
sundarga1@utrc.utc.com

**Anthony Yezzi**
School of Electrical & Computer Engineering
Georgia Institute of Technology, Atlanta, GA 30332
ayezzi@ece.gatech.edu

## Abstract

We consider the optimization of cost functionals on infinite dimensional manifolds and derive a variational approach to accelerated methods on manifolds. We demonstrate the methodology on the infinite-dimensional manifold of diffeomorphisms, motivated by optical flow problems in computer vision. We build on a variational approach to accelerated optimization in finite dimensions, and generalize that approach to infinite dimensional manifolds. We derive the continuum evolution equations, which are partial differential equations (PDE), and relate them to mechanical principles. A particular case of our approach can be viewed as a generalization of the $L^2$ optimal mass transport problem. Our approach evolves an infinite number of particles endowed with mass, represented as a mass density. The density evolves with the optimization variable, and endows the particles with dynamics. This is different than current accelerated methods where only a single particle moves and hence the dynamics does not depend on mass. We derive theory and the PDEs for acceleration, and illustrate the behavior of this new scheme.

## 1 Introduction

Accelerated optimization methods have gained wide interest within the machine learning and optimization communities (e.g., [1, 2, 3, 4, 5, 6, 7, 8, 9, 10, 11, 12, 13]). They are known for optimal convergence rates among schemes that use only gradient (first order) information in the convex case. In the non-convex case, they appear to provide robustness to shallow local minima. The intuitive idea is that by considering a particle with mass that moves in an energy landscape, the particle will gain momentum and surpass shallow local minimum and settle in in deeper and wider local extrema. These methods have so far have only been used in finite dimensional optimization problems. In this paper, we consider the generalization of these methods to infinite dimensional manifolds. We are motivated by applications in computer vision, e.g., segmentation, 3D reconstruction, and optical flow. In these problems, optimization is over infinite dimensional manifolds (e.g., curves, surfaces, mappings). Recently there has been interest within machine learning in optimization on finite dimensional manifolds, such as matrix groups, e.g., [14, 15, 16], in a non-variational framework, and the methodology presented here can be adapted to those problems as well.

Recent work [17] has shown that the continuum limit of accelerated methods may be formulated with variational principles. The resulting optimal continuum path is defined by an ordinary differential equation (ODE), which when discretized appropriately yields Nesterov's method [13]. The optimization problem is an action integral, which integrates the Bregman Lagrangian over paths. The Bregman Lagrangian consists of a generalization of kinetic and potential energies. The kinetic energy is defined using the Bregman divergence; the potential energy is the cost function that is to be optimized. We build on the approach of [17] by formulating accelerated optimization with an action integral, but we generalize that approach to manifolds. Our approach is general for manifolds, but we

illustrate the idea here for the case of the infinite dimensional manifold of diffeomorphisms of $\mathbb{R}^n$. To do this, we forgo the Bregman Lagrangian framework in [17] since that assumes that the variable over which one optimizes is embedded in $\mathbb{R}^n$, which is not the case for infinite dimensional manifolds. Instead, we adopt the formulation of action integrals in physics [18, 19], where kinetic energies that are defined through Riemannian metrics, which allows generalization beyond Euclidean geometries.

Our contributions are specifically: **1**. We present a novel variational approach to accelerated optimization on manifolds. **2**. We adapt our approach to accelerated optimization on the infinite dimensional manifold diffeomorphisms, i.e., smooth invertible mappings. **3**. We introduce a Riemannian metric for the purpose of acceleration on diffeomorphisms, which defines the kinetic energy of a mass distribution. The metric is the same one in the fluid mechanics formulation of the $L^2$ mass transport problem [20]. **4**. We derive the PDE for accelerated optimization of any cost functional defined on diffeomorphisms, and relate it to fluid mechanics principles. **5**. We present a numerical discretization, which requires entropy schemes [21], and show the advantage over gradient descent.

## 1.1 Related Work

**Optimal Mass Transport**: Our work relates to the *optimal mass transport* problem (e.g., [22, 23, 20, 24]). In this problem, two probability densities $\rho_0, \rho_1$ in $\mathbb{R}^n$ are given, and the goal is to find a transformation $M : \mathbb{R}^n \to \mathbb{R}^n$ so that the pushforward of $\rho_0$ by $M$ results in $\rho_1$ such that $M$ has minimal cost. The cost is defined as $\int_{\mathbb{R}^n} |M(x) - x|^p \rho_0(x) \, \mathrm{d}x$ where $p \geq 1$. The value of the minimum cost is called the $L^p$ Wasserstein distance. For $p = 2$, [20] has shown that mass transport can be formulated as a fluid mechanics problem. In particular, the Wasserstein distance can be formulated as a distance arising from a Riemannian metric on the space of probability densities. The tangent space consists of vector fields that infinitesimally displace the density, and the metric is the kinetic energy of the mass distribution as it is displaced by the velocity, given by $\int_{\mathbb{R}^n} \frac{1}{2} \rho(x) |v(x)|^2 \, \mathrm{d}x$. Optimal mass transport computes the optimal path that minimizes the integral of kinetic energy.

In our work, we minimize a potential on the manifold of diffeomorphisms, with acceleration. We associate a mass density in $\mathbb{R}^n$ to the diffeomorphism that, as we optimize the potential, moves in $\mathbb{R}^n$ via a push-forward of the diffeomorphism. This evolution arises as the stationary condition of the action integral, i.e., the difference of the kinetic and potential energies. One choice of kinetic energy that we explore is the $L^2$ Riemannian mass transport metric. The main difference of our approach is that we compute stationary paths of the path integral of kinetic minus *potential* energies.

**Diffeomorphic Registration**: Our work relates to diffeomorphic image registration [25, 26], where the goal is to compute registration (pixel-wise correspondence) between images as diffeomorphisms. The optimization problem is formed on a path of velocity fields, which generates a diffeomorphism. The goal is to minimize $\int_0^1 \|v\|^2 \, \mathrm{d}t$ where $v$ is a time varying vector field, and the optimization is subject to the constraint that the mapping $\phi$ maps one image to the other, i.e., $I_1 = I_0 \circ \phi^{-1}$. This minimizes an action integral where the action contains only a kinetic energy. The norm is a Sobolev norm to ensure that the generated diffeomorphism is smooth. The problem is solved by a Sobolev gradient descent (see also [27, 28, 29]) on the space of paths, which gives a geodesic.

Our framework instead uses accelerated gradient descent. Like [25, 26], it is derived from an action integral, but the action has both a kinetic energy and a *potential* energy. In our work, one choice of kinetic energy is an $L^2$ metric weighted by *mass* rather than a Sobolev metric. One of our motivations in this work is to get regularizing effects of Sobolev norms without using Sobolev norms, since that requires inverting differential operators in the optimization, which can be computationally expensive. Our approach allows us to generate diffeomorphisms without using Sobolev norms.

**Optical Flow**: The purpose of this paper is to derive the variational framework for acceleration on manifolds, in particular diffeomorphisms. To demonstrate our ideas numerically, we show gains over gradient descent on one possible application - variational optical flow (e.g., [30, 31, 32, 33, 34, 29]). Optical flow, i.e., determining pixel-wise correspondence between images, is fundamental to computer vision and remains a challenge to solve, due to its non-convexity. The dominant general approach to these problems involves iterative linearization of the cost functional around the current accumulated optical flow and its solution for an incremental displacement. This approach, in conjunction with image pyramids, has been successful in many cases, but the there are still many cases, e.g., large displacement of thin structures where it fails.

**Accelerated Optimization in Infinite Dimensions**: We present general methodology for accelerated optimization on infinite-dimensional manifolds, and illustrate it on diffeomorphisms (see [35] for an extended version of this paper). The case of the manifold of curves and surfaces is considered in [36]. Convergence rates of the resulting PDEs are not explored in this paper. However, in [37], we rigorously analyze stability conditions and time step restrictions for discretizations of accelerated PDEs and show that they are significantly more generous compared to gradient descent.

## 2 Background for Accelerated Optimization on Manifolds

**Differential Geometry**: We review basic manifold theory [38]. A *manifold $M$* is a space in which every point $p \in M$ has a (invertible) mapping $f_p$ from a neighborhood of $p$ to a *model space* that is a linear normed vector space, and has an additional compatibility condition that if the neighborhoods for $p$ and $q$ overlap then the mapping $f_p \circ f_q^{-1}$ is differentiable. Intuitively, a manifold is a space that locally appears flat. The manifold may be finite or infinite dimensional when the model spaces are finite or infinite dimensional, respectively. The *tangent space* at a point $p \in M$ is the equivalence class, $[\gamma]$, of curves $\gamma : [0, 1] \to M$ under the equivalence that $\gamma(0) = p$ and $(f_p \circ \gamma)'(0)$ are the same for each curve $\gamma \in [\gamma]$. Intuitively, these are the set of possible directions of movement at the point $p$ on the manifold. The *tangent bundle*, denoted $TM$, is $TM = \{(p, v) : p \in M, v \in T_p M\}$, i.e., the space formed from the collection of all points and tangent spaces.

A *Riemannian manifold* is a manifold that has an inner product (called the *metric*) that exists on each tangent space $T_p M$, and smoothly varies on $M$. A Riemannian manifold allows one to formally define the lengths of curves $\gamma : [-1, 1] \to M$ on the manifold. This allows one to construct paths of critical length, called *geodesics*, a generalization of a path of constant velocity. The Riemannian metric allows one to define *gradients* of functions $g : M \to \mathbb{R}$ defined on the manifold: the gradient $\nabla g(p) \in T_p M$ is defined to be the vector that satisfies $\frac{\mathrm{d}}{\mathrm{d}\varepsilon} g(\gamma(\varepsilon))|_{\varepsilon=0} = \langle \nabla g(p), \gamma'(0) \rangle$, where $\gamma(0) = p$, the left hand side is the directional derivative and the right hand side is the inner product.

**Mechanics on Manifolds**: We now review some of the formalism of classical mechanics on manifolds [18, 19]. The subject of mechanics describes the principles governing the evolution of a particle that moves on a manifold $M$. The equations governing a particle are Newton's laws. There are two viewpoints in mechanics, namely the *Lagrangian* and *Hamiltonian* viewpoints, which formulate more general principles to derive Newton's equations. In this paper, we use the Lagrangian formulation to derive equations of motion for accelerated optimization on the manifold of diffeomorphisms. Lagrangian mechanics obtains equations of motion through *variational principles*, which makes it easier to generalize Newton's laws beyond simple particle systems in $\mathbb{R}^3$, especially to the case of manifolds. In Lagrangian mechanics, one starts with a Lagrangian $L : TM \to \mathbb{R}$ where $M$ is a Riemannian manifold. One says that a curve $\gamma : [-1, 1] \to M$ is *a motion in a Lagrangian system* with Lagrangian $L$ if it is an extremal of $A = \int L(\gamma(t), \dot{\gamma}(t)) \, \mathrm{d}t$. The previous integral is called an *action integral*. *Hamilton's principle of stationary action* states that the motion in the Lagrangian system satisfies the condition that $\delta A = 0$, where $\delta$ denotes the variation, for *all* variations of $A$ induced by variations of the path $\gamma$ that keep endpoints fixed. The variation is defined as $\delta A := \frac{\mathrm{d}}{\mathrm{d}s} A(\tilde{\gamma}(t, s))|_{s=0}$ where $\tilde{\gamma} : [-1, 1]^2 \to M$ is a smooth family of curves (a variation of $\gamma$) on the manifold such that $\tilde{\gamma}(t, 0) = \gamma(t)$. The stationary conditions give rise to what is known as *Lagrange's* equations. A *natural Lagrangian* has the special form $L = T - U$ where $T : TM \to \mathbb{R}^+$ is the *kinetic energy* and $U : M \to \mathbb{R}$ is the *potential energy*. The kinetic energy is defined as $T(v) = \frac{1}{2} \langle v, v \rangle$ where $\langle \cdot, \cdot \rangle$ is the Riemannian metric. In the case that one has a particle system in $\mathbb{R}^3$, i.e., a collection of particles with masses $m_i$, in a natural Lagrangian system, one can show that Hamilton's principle of stationary action is equivalent to Newton's law of motion, i.e., that $\frac{\mathrm{d}}{\mathrm{d}t}(m_i \dot{r}_i) = -\frac{\partial U}{\partial r_i}$ where $r_i$ is the trajectory of the $i^{\mathrm{th}}$ particle, and $\dot{r}_i$ is the velocity. This states that mass times acceleration is the force, which is given by minus the derivative of the potential in a conservative system. Thus, Hamilton's principle is more general and allows us to more easily derive equations of motion for more general systems, in particular those on manifolds.

In this paper, we will consider *Lagrangian non-autonomous systems* where the Lagrangian is also an explicit function of time $t$, i.e., $L : TM \times \mathbb{R} \to \mathbb{R}$. In particular, the kinetic and potential energies can both be explicit functions of time: $T : TM \times \mathbb{R} \to \mathbb{R}$ and $U : M \times \mathbb{R} \to \mathbb{R}$. Autonomous systems have an *energy conservation property* and do not converge; for instance, one can think of a

moving pendulum with no friction, which oscillates forever. Since we wish to minimize an objective functional, we want the system to converge and Lagrangian non-autonomous systems allow for this.

**Variational Approach to Accelerated Optimization in** $\mathbb{R}^n$: We review the variational formulation of accelerated gradient descent by [17]. This approach is based on the Bregman Lagrangian:

$$L(X, V, t) = e^{a(t)+\gamma(t)} \left[ d(X + e^{-a(t)}V, X) - e^{b(t)}U(X) \right],$$

where the potential energy $U$ represents the cost to be minimized, and $d(y, x) = h(y) - h(x) - \nabla h(x) \cdot (y - x)$ where $h$ is a given convex function. In the Euclidean case, where $h(x) = \frac{1}{2}|x|^2$, $d(y, x) = \frac{1}{2}|y - x|^2$, this simplifies to

$$L(X, V, t) = e^{\gamma(t)} \left[ e^{-a(t)}|V|^2/2 - e^{a(t)+b(t)}U(X) \right],$$

where $T = \frac{1}{2}|V|^2$ is the kinetic energy of a unit mass particle in $\mathbb{R}^n$. Nesterov's methods [13, 39, 40, 12, 41, 11] belong to a subfamily of Bregman Lagrangians with various choices of $a, b, \gamma$.

The Bregman Lagrangian assumes that the underlying manifold is a subset of $\mathbb{R}^n$, which is not true of many manifolds including diffeomorphisms[1]. Therefore, we use the mechanics formulation, which provides a formalism for general metrics though the Riemannian distance.

# 3 Accelerated Optimization on Manifolds Applied to Diffeomorphisms

In this section, we use the mechanics on manifolds to generalize accelerated optimization to infinite dimensional manifolds, in particular the manifold of diffeomorphisms of $\mathbb{R}^n$ for general $n$. Diffeomorphisms, denoted $\text{Diff}(\mathbb{R}^n)$, are smooth mappings $\phi : \mathbb{R}^n \to \mathbb{R}^n$ whose inverse exists and is also smooth. The cost functional (the potential) is denoted $U(\phi)$ and our framework applies to any potential. In the first sub-section, we give the formulation and evolution equations for the case of acceleration without energy dissipation, i.e., the Lagrangian is autonomous, since it is relevant for the case of energy dissipation. In the second sub-section, we formulate the dissipative case, which generalizes [17] to diffeomorphisms. All proofs are found in supplementary material.

## 3.1 Acceleration Without Energy Dissipation

**Formulation of the Action Integral**: To formulate the action, we define the kinetic energy $T$ on the space of diffeomorphisms, which is defined on the tangent space, denoted $T_\phi \text{Diff}(\mathbb{R}^n)$, to $\text{Diff}(\mathbb{R}^n)$ at a diffeomorphism $\phi$. The tangent space at $\phi$ is the set of perturbations $v$ of $\phi$ that preserve the diffeomorphism property, i.e., for all small $\varepsilon$, $\phi + \varepsilon v$ is a diffeomorphism. One can show that

$$T_\phi \text{Diff}(\mathbb{R}^n) = \{v : \phi(\mathbb{R}^n) \to \mathbb{R}^n \ : \ v \text{ is in a Sobolev space }\}, \qquad (1)$$

which is a set of smooth vector fields on $\phi(\mathbb{R}^n)$ in which the vector field at each point $\phi(x)$ displaces $\phi(x)$ infinitesimally. Since $\phi$ is a diffeomorphism, we have that $\phi(\mathbb{R}^n) = \mathbb{R}^n$. However, we write $v : \phi(\mathbb{R}^n) \to \mathbb{R}^n$ to emphasize that the velocity fields in the tangent space are defined on the deformed domain, so that $v$ is a Eulerian velocity.

We note a result from [42], which will be the basis of our derivation of accelerated optimization on $\text{Diff}(\mathbb{R}^n)$. Any (orientable) diffeomorphism may be generated by integrating a time-varying smooth vector field over time, i.e.,

$$\partial_t \phi_t(x) = v_t(\phi_t(x)), \quad x \in \mathbb{R}^n, \qquad (2)$$

where $\partial_t$ denotes partial derivative with respect to $t$, $\phi_t$ denotes a time varying family of diffeomorphisms evaluated at the time $t$, and $v_t$ is a time varying collection of vector fields evaluated at time $t$. The path $t \to \phi_t(x)$ for a fixed $x$ represents a trajectory of a particle starting at $x$.

The space on which the kinetic energy is defined is now clear, but one more ingredient is needed before we can define the kinetic energy. Any accelerated method will need a notion of *mass*, as a mass-less ball will not accelerate. We imagine that an infinite number of particles densely distributed in $\mathbb{R}^n$ with mass exist and are displaced by the velocity field $v$ at every point. We represent the mass

distribution with a *mass density* $\rho : \mathbb{R}^n \to \mathbb{R}^+$, which is the mass divided by volume as the volume shrinks. During the evolution to optimize the potential $U$, the particles are displaced continuously and thus the density of these particles will in general change over time. We will assume that the system of particles in $\mathbb{R}^n$ is closed and so we impose *mass preservation*, i.e., $\int_{\mathbb{R}^n} \rho(x)\,\mathrm{d}x = 1$. The evolution of a time varying density $\rho_t$ as it is deformed by a time varying velocity is given by the *continuity equation*, which is a local form of the conservation of mass, given by

$$\partial_t \rho(x) + \mathrm{div}\,(\rho(x)v(x)) = 0, \quad x \in \mathbb{R}^n \tag{3}$$

where div () denotes the divergence operator, given by $\mathrm{div}\,(F) = \sum_{i=}^n \partial_{x_i} F^i$ where $\partial_{x_i}$ is the partial with respect to the $i^{\text{th}}$ coordinate and $F^i$ is the $i^{\text{th}}$ component of the vector field.

We now have the ingredients to define the kinetic energy. We present one choice to illustrate the idea of accelerated optimization. We define the kinetic energy as

$$T(v) = \int_{\phi(\mathbb{R}^n)} \frac{1}{2}\rho(x)|v(x)|^2\,\mathrm{d}x, \tag{4}$$

which matches the definition of the kinetic energy of a system of particles in physics.

We define the action integral, which is defined on *paths* on $\mathrm{Diff}(\mathbb{R}^n)$. A path of diffeomorphisms is $\phi : [0, \infty) \times \mathbb{R}^n \to \mathbb{R}^n$. We denote the diffeomorphism at a time $t$ as $\phi_t$. Since diffeomorphisms are generated by velocity fields, we can define the action on paths of velocity fields. A path of velocity fields is given by $v : [0, \infty) \times \mathbb{R}^n \to \mathbb{R}^n$. The kinetic energy is dependent on the mass density, thus, a path of densities $\rho : [0, \infty) \times \mathbb{R}^n \to \mathbb{R}^+$ is required, which represents the mass distribution as it is deformed. The action integral is

$$A = \int [T(v_t) - U(\phi_t)]\,\mathrm{d}t, \tag{5}$$

where the integral is over time. The action is implicitly a function of three paths, i.e., $v_t$, $\phi_t$ and $\rho_t$. These paths are coupled as $\phi_t$ depends on $v_t$ through (2), and $\rho_t$ depends on $v_t$ through (3).

**Stationary Conditions for the Action**: We treat the computation of the stationary conditions of the action as a constrained optimization problem with respect to the two aforementioned constraints. To do this, it is easier to formulate the action in terms of the path of the inverse diffeomorphisms $\phi_t^{-1}$, which we will call $\psi_t$. This is because the non-linear PDE constraint (2) can be equivalently reformulated as the following linear transport PDE in the inverse mappings:

$$\partial_t \psi_t(x) + [D\psi_t(x)]v_t(x) = 0, \quad x \in \mathbb{R}^n \tag{6}$$

where $D$ denotes the derivative (Jacobian) operator. To derive the stationary conditions with respect to the constraints, we use the method of Lagrange multipliers. We denote by $\lambda : [0, \infty) \times \mathbb{R}^n \to \mathbb{R}^n$ the Lagrange multiplier according to (6). We denote $\mu : [0, \infty) \times \mathbb{R}^n \to \mathbb{R}$ as the Lagrange multiplier for the continuity equation (3). Because we would like to be able to have possibly discontinuous solutions of the continuity equation, we formulate it in its weak form by multiplying the constraint by the Lagrange multiplier and integrating by parts thereby removing the derivatives on possibly discontinuous $\rho$. This gives the action integral with Lagrange multipliers as

$$A = \int [T(v) - U(\phi)]\,\mathrm{d}t + \int\int_{\mathbb{R}^n} \lambda^T[\partial_t\psi + (D\psi)v]\,\mathrm{d}x\,\mathrm{d}t - \int\int_{\mathbb{R}^n} [\partial_t\mu + \nabla\mu \cdot v]\,\rho\,\mathrm{d}x\,\mathrm{d}t, \tag{7}$$

where we have omitted the subscripts to avoid cluttering the notation. Notice that the potential $U$ is now a function of $\psi$, and the action depends on $\rho, \psi, v$ and the Lagrange multipliers $\mu, \lambda$.

We now compute variations of $A$ as we perturb the paths by variations $\delta\rho$, $\delta v$ and $\delta\phi$ along the paths. The variation with respect to $\rho$ is defined as $\delta A \cdot \delta\rho = \frac{\mathrm{d}}{\mathrm{d}\varepsilon}A(\rho + \varepsilon\delta\rho, v, \psi)\big|_{\varepsilon=0}$, and the other variations are defined in a similar fashion. This results in the following theorem.

**Theorem 3.1** (Evolution Equations for the Path of Least Action). *The stationary conditions for the path of the action integral* (5) *subject to the constraints* (2) *on the mapping and the continuity equation* (3) *are given by the forward evolution equation*

$$\rho\frac{Dv}{Dt} = -\nabla U(\phi), \quad or \quad \partial_t v = -(Dv)v - \frac{1}{\rho}\nabla U(\phi), \tag{8}$$

*where* $\frac{Df}{Dt} := \partial_t f + (Df)v$ *is the material derivative. The previous equation describes the velocity evolution. The forward evolution equation for the diffeomorphism is given by* (2)*, that of its inverse mapping is given by* (6)*, and the forward evolution of its density is given by* (3)*.*

The first equation in (8) is an analogue of Newton's equations. Indeed, the equation says the time rate of change of velocity along trajectories generated by the velocity field multiplied by density is equal to minus the gradient of the potential, which is Newton's 2nd law.

**Viscosity Solution and Regularity**: The evolution equations given by Theorem 3.1 maintain the mapping $\phi_t$ as a diffeomorphism. This is because we define the solution as the *viscosity solution* (e.g., [43, 44, 21]). The viscosity solution is defined as follows. Define

$$\partial_t v_\varepsilon = -(Dv_\varepsilon)v_\varepsilon + \varepsilon \Delta v_\varepsilon - \rho^{-1}\nabla U(\phi), \qquad (9)$$

where $\Delta$ denotes the spatial Laplacian, which is a smoothing operator. This leads to a smooth ($C^\infty$) solution due to smoothing properties of the Laplacian. The viscosity solution is $v = \lim_{\varepsilon \to 0} v_\varepsilon$. We approximate the effects with small $\varepsilon$ by using entropy conditions in our numerical implementation. Since the velocity is smooth ($C^\infty$), the integral of a smooth vector field is a diffeomorphism [42].

## 3.2 Acceleration with Energy Dissipation

We now present the case of a non-autonomous Lagrangian. We consider time varying scalar functions $a, b : [0, \infty) \to \mathbb{R}^+$, and define the action integral as follows:

$$A = \int [a_t T(v_t) - b_t U(\phi_t)] \, \mathrm{d}t, \qquad (10)$$

where $a_t, b_t$ denote the values of the scalar at time $t$. It can be shown that the stationary conditions result in the following evolution:

**Theorem 3.2** (Evolution Equations for the Path of Least Action). *The stationary conditions for the path of the action integral* (10) *subject to the constraints* (2) *on the mapping and the continuity equation* (3) *are given by the forward evolution equation*

$$a\partial_t v + a(Dv)v + (\partial_t a)v = -(b/\rho)\nabla U(\phi), \qquad (11)$$

*which describes the evolution of the velocity. The same evolution equations as Theorem 3.1 for the mappings* (2) *and* (6)*, and density* (3) *hold.*

If we consider the case $a_t = e^{\gamma_t - \alpha_t}, b_t = e^{\alpha_t + \beta_t + \gamma_t}$ where $\alpha_t = \log p - \log t, \beta_t = p \log t + \log C, \gamma_t = p \log t$, with $p = 2$, which was considered in [17] as the continuum limit of Nesterov's original scheme in finite dimensions, then we arrive at the following evolution equation:

$$\partial_t v = -(3/t)v - (Dv)v - (1/\rho)\nabla U(\phi). \qquad (12)$$

This evolution equation is the same as the evolution equations for the non-dissipative case (8), except for the term $-(3/t)v$. One can interpret the latter term as a frictional dissipative term.

# 4 Experiments

We now show empirical evidence to illustrate the behavior of our accelerated optimization by comparing it to gradient descent. For illustration, we consider:

$$U(\phi) = \frac{1}{2}\int_{\mathbb{R}^n} |I_1(\phi(x)) - I_0(x)|^2 \, \mathrm{d}x + \frac{1}{2}\alpha \int_{\mathbb{R}^n} |\nabla(\phi(x) - x)|^2 \, \mathrm{d}x, \qquad (13)$$

where $\alpha > 0$ is a weight, and $I_0, I_1$ are images, which is the classical Horn & Schunck energy. The first term is the data fidelity which measures how close $\phi$ deforms $I_1$ back to $I_0$ through the squared norm, and the second term penalizes non-smoothness of the displacement field, given by $\phi(x) - x$.

We compare to standard (Riemannian $L^2$) gradient descent to illustrate how much one can gain by incorporating acceleration, which requires little additional effort over gradient descent. Over gradient descent, acceleration requires only to update the velocity by the velocity evolution in the previous section, and the density evolution. Both these evolutions are cheap to compute since they only involve local updates. We discretize using forward Euler and entropy conditions (see Supplementary for details), and choose the step size to satisfy CFL conditions. For gradient descent we choose $\Delta t < 1/(4\alpha)$; for accelerated gradient descent we have the additional evolution of the velocity (12), and our numerical scheme has CFL condition $\Delta t < 1/(4\alpha \cdot \max_{x \in \Omega}\{|v(x)|, |Dv(x)|\})$. The initialization is $\phi(x) = \psi(x) = x, v(x) = 0$, and $\rho(x) = 1/|\Omega|$ where $|\Omega|$ is the area the image.

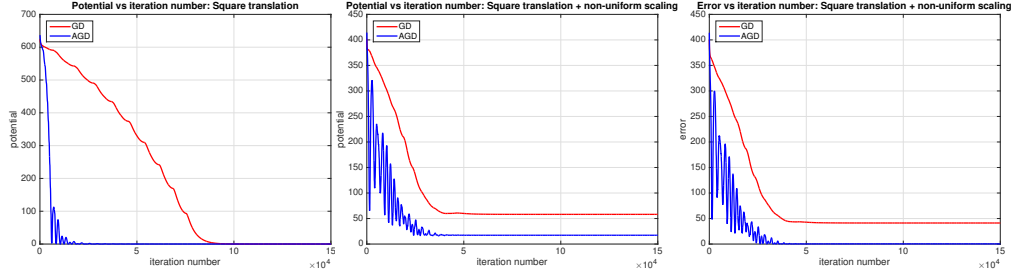

Figure 1: **Convergence Comparison**: Two images are registered, each are binary images. [Left]: The two images are a square and a translation of it. [Middle and right]: The first image is a square and the second image is a translated and non-uniformly scaled version of the square in the first image. [Left and middle]: The cost functional to be minimized versus the iteration number is shown for both gradient descent (GD) and accelerated gradient descent (AGD). [Right]: The image reconstruction error: $\|I_1 \circ \phi - I_0\|$ in the non-uniformly scaled squares is shown.

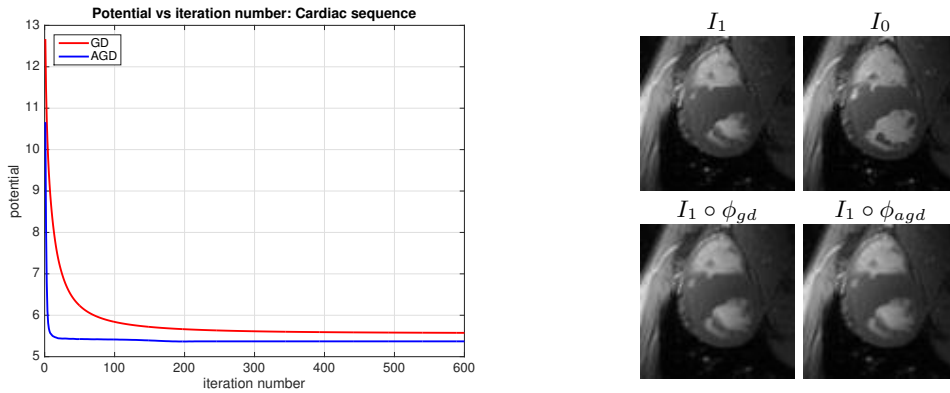

Figure 2: **Convergence Comparison**: Two MR cardiac images are registered. [Left]: A plot of the potential versus the iteration number in the minimization using gradient descent (GD) and accelerated gradient descent (AGD). [Right]: The original images and the back-warped images using the recovered diffeomorphisms. Note that $I_1 \circ \phi$ should appear close to $I_0$.

**Convergence analysis**: In this experiment, the images are two white squares against a black background. The sizes of the squares are $50 \times 50$ pixels wide, and the square (of size $20 \times 20$) in the first image is translated by 10 pixels to form the second image. Small images are chosen due to the fact gradient descent is impractically slow for larger sized images (e.g., $256 \times 256$). Figure 1 (left figure) shows the plot of the potential energy (13) of both gradient descent and accelerated gradient descent. Here $\alpha = 5$. Accelerated gradient descent very quickly accelerates to a global minimum, surpasses the global minimum and then oscillates until the friction term slows it down and then it converges very quickly. Gradient descent slowly decreases the energy and eventually converges.

We now repeat the same experiment, but with different images. We choose the images again to be $50 \times 50$. The first image has a square that is $17 \times 17$ and the second image has a rectangle of size $20 \times 14$ and is translated by 8 pixels. We choose $\alpha = 2$. A plot of results is shown in Figure 1 (middle). Again accelerated gradient descent accelerates very quickly at the start, then oscillates and the oscillations die down and then it converges. The potential is not zero as the flow is not a translation and thus the regularity term is non-zero. Gradient descent converges faster than the case of translation due to smaller $\alpha$ and thus larger step size. However, it is stuck at a high energy configuration. Gradient descent has not fully converged. We verify this by plotting the first term of the potential, which is zero for accelerated gradient descent at convergence, indicating a perfect match. Gradient descent has an error of 50, indicating the flow does not fully warp $I_1$ to $I_0$.

We repeat the experiment with cardiac MR images. The image transformation is a general diffeomorphism. We choose $\alpha = 0.02$. A plot of potential versus iterations for both methods is shown in Figure 2. Convergence is quicker for AGD, though both schemes converge to a similar solution.

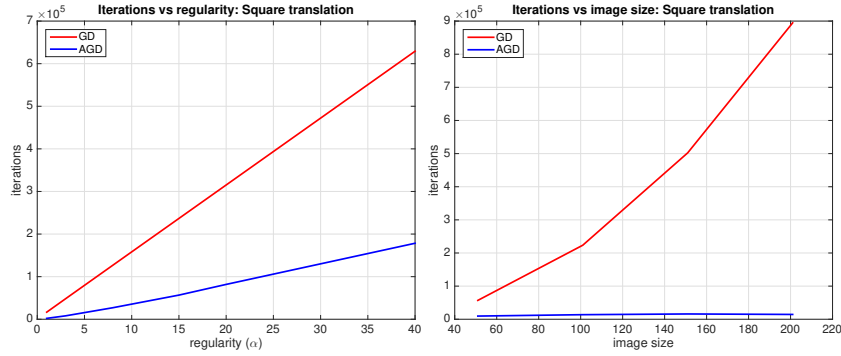

Figure 3: [Left]: **Convergence Comparison as a Function of Regularity**: Two binary images are registered with varying amounts of regularization $\alpha$ for gradient descent (GD) and accelerated gradient descent (AGD). [Right]: **Convergence Comparison as a Function of Image Size**: We vary the size (height and width) of the image and compare GD with AGD.

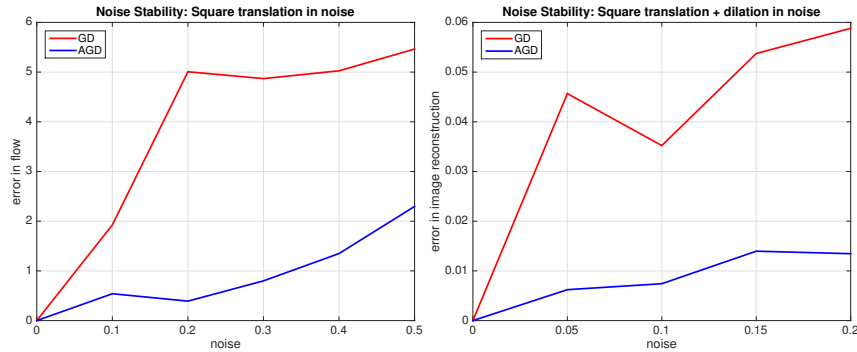

Figure 4: **Analysis of Stability to Noise**: We register noisy images with varying amounts of noise. We plot the error in the recovered flow of both GD and AGD versus the level of noise. [Left]: The first image is formed from a square and the second image is the same square but translated. [Right]: The first image is a square and the second image is the non-uniformly scaled and translated square.

**Convergence analysis versus parameter settings**: We now analyze the convergence of accelerated gradient descent and gradient descent as a function of the regularity $\alpha$ and the image size. First, we analyze an image pair of size $50 \times 50$ in which one image has a square of size $16 \times 16$ and the other image is the same square translated by 7 pixels. We vary $\alpha$ and analyze the convergence. In the left plot of Figure 3, we show the number of iterations until convergence versus the regularity $\alpha$. As $\alpha$ increases, the number of iterations for both gradient descent and accelerated gradient descent increase. However, the number of iterations for accelerated gradient descent grows more slowly. In all cases, the flow achieves the ground truth flow. Next, we analyze the number of convergence iterations versus the image size. We consider binary images with squares of size $16 \times 16$ and translated by 7 pixels. However, we vary the image size from $50 \times 50$ to $200 \times 200$. We fix $\alpha = 8$. We show the number of iterations to convergence versus the image size in the right of Figure 3. Gradient descent is impractically slow for all the sizes considered, and the number of iterations quickly increases with image size. Accelerated gradient descent appears to have little growth with respect to the image size.

**Analysis of Robustness to Noise**: We simulate undesirable local minima by using salt and pepper noise. We consider images of size $50 \times 50$. We fix $\alpha = 1$ and vary the noise level. One could increase $\alpha$ to increase robustness to noise; however, we are interested in understanding the robustness to noise of the optimization algorithms. We consider a square of size $16 \times 16$ in a binary image and the same square translated by 4 pixels in the second image. We plot the error in the flow versus the noise level. The result is shown in the left plot of Figure 4. This shows that accelerated gradient descent degrades much slower than gradient descent. We repeat the experiment with different images, one with a square of size $15 \times 15$ and a rectangle that is $20 \times 10$ and translated by 5 pixels. The result is plotted in the right of Figure 4. A similar trend as in the previous experiment is observed.

**Acknowledgments**

This research was partially funded by ARO W911NF-18-1-0281 and NSF CCF-1526848.

## Footnotes

[1]The Bregman distance can be generalized to manifolds using the exponential and logarithmic maps. However, for many manifolds, including diffeomorphisms, computing these maps require solving an optimization problem.

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
