[Supplementary Material]



# Variational PDEs for Acceleration on Manifolds and Application to Diffeomorphisms: Supplementary Material

Anonymous NIPS Submission

## I. EXPERIMENTS: SOME VISUALIZATION OF RESULTS

We include the visualizations for the 'Analysis of Robustness' to noise experiments in the manuscript. Figure 1 corresponds to the experiment in Figure 4 (left) in the manuscript, and Figure 2 corresponds to the experiment in Figure 4 (right) in the manuscript.

## II. DERIVATIONS AND PROOFS OF THEOREMS

### A. Functional Gradients

We present first the definition of functional gradients so that we can define the gradient of the potential.

**Definition 1** (Functional Gradients). *Let $U : Diff(\mathbb{R}^n) \to \mathbb{R}$. The gradient (or functional derivative) of $U$ with respect to $\phi \in Diff(\mathbb{R}^n)$, denoted $\nabla U(\phi)$, is defined as the $\nabla U(\phi) \in T_\phi Diff(\mathbb{R}^n)$ that satisfies*

$$\delta U(\phi) \cdot v = \int_{\phi(\mathbb{R}^n)} \nabla U(\phi)(x) \cdot v(x) \, \mathrm{d}x \tag{1}$$

*for all $v \in T_\phi Diff(\mathbb{R}^n)$. The left hand side is the directional derivative and is defined as*

$$\delta U(\phi) \cdot v := \left. \frac{\mathrm{d}}{\mathrm{d}\varepsilon} U(\phi + \varepsilon v) \right|_{\varepsilon=0} . \tag{2}$$

*Note that $(\phi + \varepsilon v)(x) = \phi(x) + \varepsilon v(\phi(x))$ for $x \in \mathbb{R}^n$.*

We now show the computation of the gradient for the illustrative potential used in this paper. First, let us consider the data term $U_1(\phi) = \int_{\mathbb{R}^n} |I_1(\phi(x)) - I_0(x)|^2 \, \mathrm{d}x$ then

$$\delta U_1(\phi) \cdot \delta\phi = \int_{\mathbb{R}^n} 2(I_1(\phi(x)) - I_0(x)) DI_1(\phi(x)) \widehat{\delta\phi}(x) \, \mathrm{d}x = \int_{\phi(\mathbb{R}^n)} 2(I_1(x) - I_0(\psi(x))) DI_1(x) \delta\phi(x) \det \nabla\psi(x) \, \mathrm{d}x,$$

Fig. 1. **Visual Comparison on Square Translation in Noise Experiment**. The above show the visual results of the noise robustness experiment. For each row group of images: the two original images, the warped image by gradient descent, and the warped image by accelerated gradient descent. The last two images should resemble the second if the registration is correct.

Fig. 2. **Visual Comparison on Square Non-Uniform Scaling and Translation in Noise Experiment**. The above show the visual results of the noise robustness experiment. For each row group of images: the two original images, the warped image by gradient descent, and the warped image by accelerated gradient descent. The last two images should resemble the second if the registration is correct.

where $\widehat{\delta\phi} = \delta\phi \circ \phi$, $\psi = \phi^{-1}$ and we have performed a change of variables. Thus, $\nabla U_1 = 2\nabla I_1(I_1 - I_0 \circ \psi) \det \nabla\psi$. Now consider the regularity term $U_2(\phi) = \int_{\mathbb{R}^n} |\nabla(\phi(x) - x)|^2 \, \mathrm{d}x$, then

$$\delta U(\phi) = 2 \int_{\mathbb{R}^n} \mathrm{tr}\left(\nabla(\phi(x) - \mathrm{id})^T \nabla\widehat{\delta\phi}(x)\right) \, \mathrm{d}x = -\int_{\mathbb{R}^n} \Delta\phi(x)^T \delta\phi(x) \, \mathrm{d}x = \int_{\Omega} (\Delta\phi)(\psi(x))^T \delta\phi(x) \det \nabla\psi(x) \, \mathrm{d}x.$$

Note that in integration by parts, the boundary term vanishes since we assume that $\phi(x) = x$ as $|x| \to \infty$. Thus, $\nabla U_2 = (\Delta\phi) \circ \psi \det \nabla\psi$.

### B. Stationary Conditions

**Lemma 1** (Stationary Condition for the Mapping). *The stationary condition of the action defined in Eqn 7 (manuscript) for the mapping is*

$$\partial_t \lambda + div\left(v\lambda^T\right)^T = (\nabla\psi)^{-1}\nabla U(\phi). \tag{3}$$

*Proof.* We compute the variation of $A$ (defined in Eqn 7 manuscript) with respect to the mapping $\phi$. The only terms in the action that depend on the mapping are $U$ and the Lagrange multiplier term associated with the mapping. Taking the variation w.r.t the potenial term gives

$$-\int \int_{\phi(\mathbb{R}^n)} \nabla U(\phi) \cdot \delta\phi \, \mathrm{d}x \, \mathrm{d}t.$$

Now the variation with respect to the Lagrange multiplier term:

$$\int \int_{\phi(\mathbb{R}^n)} \lambda^T[\partial_t \widehat{\delta\psi} + D(\widehat{\delta\psi})v] \, \mathrm{d}x \, \mathrm{d}t = -\int \int_{\phi(\mathbb{R}^n)} [\partial_t \lambda^T + div\left(v\lambda^T\right)]\widehat{\delta\psi} \, \mathrm{d}x \, \mathrm{d}t,$$

where we have integrated by parts, the $div\left(\cdot\right)$ of a matrix means the divergence of each of the columns, resulting in a row vector, and $\widehat{\delta\psi} = \delta\psi \circ \psi$. Note that we can take the variation of $\psi(\phi(x)) = x$ to obtain

$$\widehat{\delta\psi} \circ \phi(x) + [D\psi(\phi(x))]\widehat{\delta\phi}(x) = 0,$$

or

$$\widehat{\delta\psi}(y) = -[D\psi(y)]\delta\phi(y).$$

Therefore,

$$\delta A \cdot \delta\phi = \int \int_{\phi(\mathbb{R}^n)} \left\{ (\nabla\psi)\left[\partial_t \lambda + div\left(v\lambda^T\right)^T\right] - \nabla U(\phi) \right\} \cdot \delta\phi \, \mathrm{d}x \, \mathrm{d}t. \tag{4}$$

$\square$

**Lemma 2** (Stationary Condition for the Velocity). *The stationary condition of the action in Eqn. 7 (manuscript) arising from the velocity is*

$$\rho v + (\nabla\psi)\lambda - \rho\nabla\mu = 0. \tag{5}$$

*Proof.* We compute the variation w.r.t the kinetic energy:

$$\delta T \cdot \delta v = \int_{\phi(\mathbb{R}^n)} \rho v \cdot \delta v \, \mathrm{d}x.$$

The variation of the Lagrange multiplier terms is

$$\int \int_{\phi(\mathbb{R}^n)} \lambda^T (D\psi)\delta v - \rho \nabla \mu \cdot \delta v \, \mathrm{d}x \, \mathrm{d}t = \int \int_{\phi(\mathbb{R}^n)} [(\nabla\psi)\lambda - \rho\nabla\mu] \cdot \delta v \, \mathrm{d}x \, \mathrm{d}t.$$

Therefore,

$$\delta A \cdot \delta v = \int \int_{\phi(\mathbb{R}^n)} [\rho v + (\nabla\psi)\lambda - \rho\nabla\mu] \cdot \delta v \, \mathrm{d}x \, \mathrm{d}t. \tag{6}$$

$\square$

**Lemma 3** (Stationary Condition for the Density). *The stationary condition of the action in Eqn. 7 (manuscript) arising from the velocity is*

$$\partial_t \mu + (D\mu)v = \frac{1}{2}|v|^2. \tag{7}$$

*Proof.* Note that the terms that contain the density in Eqn 7 (manuscript) are the kinetic energy and the Lagrange multiplier corresponding to the density. We see that

$$\delta A \cdot \delta \rho = \int \int_{\phi(\mathbb{R}^n)} \frac{1}{2}|v|^2 \delta\rho - (\partial_t\mu + \nabla\mu \cdot v)\delta\rho \, \mathrm{d}x \, \mathrm{d}t, \tag{8}$$

which yields the lemma. $\square$

*C. Velocity Evolution*

**Lemma 4.** *Given that $(\nabla\psi)\lambda = w$, we have that*

$$\partial_t \lambda + (D\lambda)v + \lambda div(v) = (\nabla\psi)^{-1}[\partial_t w + (Dw)v + (\nabla v)w + wdiv(v)] \tag{9}$$

*Proof.* Define the Hessian as follows:

$$[D^2\psi]_{ijk} = \partial^2_{x_i x_j}\psi^k, \quad [D^2\psi(a,b)]_k = \sum_{ij} \partial^2_{x_i x_j}\psi^k a_i b_j.$$

We compute

$$\{D[(\nabla\psi)\lambda]\}_{ij} = \partial_{x_j}[(\nabla\psi)\lambda]_i = \partial_{x_j}\sum_l \partial_{x_i}\psi^l \lambda_l = \sum_l (\partial^2_{x_j x_i}\psi^l \lambda_l) + \partial_{x_i}\psi^l \partial_{x_j}\lambda_l.$$

Therefore,

$$D[(\nabla\psi)\lambda] = D^2\psi(\cdot,\cdot) \cdot \lambda + (\nabla\psi)(D\lambda)$$

Since $D[(\nabla\psi)\lambda] = Dw$ then solving for $D\lambda$ gives

$$D\lambda = (\nabla\psi)^{-1}[Dw - D^2\psi(\cdot,\cdot) \cdot \lambda],$$

so

$$(D\lambda)v = (\nabla\psi)^{-1}[(Dw)v - D^2\psi(\cdot,v) \cdot \lambda]. \tag{10}$$

Now differentiating $(\nabla\psi)\lambda = w$ w.r.t $t$, we have

$$(\nabla\partial_t\psi)\lambda + (\nabla\psi)\partial_t\lambda = \partial_t w, \quad \text{or} \quad \partial_t\lambda = (\nabla\psi)^{-1}[\partial_t w - (\nabla\partial_t\psi)\lambda]$$

Note that $\partial_t\psi = -(D\psi)v$ so

$$\partial_t\lambda = (\nabla\psi)^{-1}\{\partial_t w + \nabla[(D\psi)v]\lambda\}. \tag{11}$$

Now computing $\nabla[(D\psi)v]$ yields

$$\{\nabla[(D\psi)v)]\}_{lk} = \partial_{x_l}\sum_i \partial_{x_i}\psi^k v^i = \sum_i \partial_{x_l}\partial_{x_i}\psi^k v^i + \partial_{x_i}\psi^k \partial_{x_l}v^i.$$

Then multiplying the above matrix by $\lambda$ gives

$$\{\nabla[(D\psi)v]\lambda\}_l = \sum_{ik} \partial_{x_l}\partial_{x_i}\psi^k v^i \lambda^k + \partial_{x_i}\psi^k \partial_{x_l}v^i \lambda^k,$$

which in matrix form is

$$\nabla[(D\psi)v)]\lambda = D^2\psi(\cdot, v) \cdot \lambda + (\nabla v)(\nabla \psi)\lambda = D^2\psi(\cdot, v) \cdot \lambda + (\nabla v)w$$

Therefore, (11) becomes

$$\partial_t \lambda = (\nabla \psi)^{-1}[\partial_t w + D^2\psi(\cdot, v) \cdot \lambda + (\nabla v)w].$$

Combining the previous with (10) and noting that $\lambda \text{div}(v) = (\nabla \psi)^{-1}w\text{div}(v)$ yields

$$\partial_t \lambda + (D\lambda)v + \lambda \text{div}(v) = (\nabla \psi)^{-1}[\partial_t w + (Dw)v + (\nabla v)w + w\text{div}(v)].$$

$\square$

**Lemma 5.** *If $w = \rho(\nabla\mu - v)$, then*

$$\partial_t w + (Dw)v + (\nabla v)w + w\,div\,(v) = -\rho[\partial_t v + (Dv)v]. \tag{12}$$

*Proof.* Differentiating $w = \rho(\nabla\mu - v)$, we have

$$\partial_t w = (\partial_t \rho)(\nabla\mu - v) + \rho(\nabla\partial_t\mu - \partial_t v)$$
$$Dw = (\nabla\mu - v)(D\rho) + \rho[D(\nabla\mu) - Dv].$$

Therefore,

$$\partial_t w + (Dw)v + (\nabla v)w + w\text{div}(v) = (\nabla\mu - v)(\partial_t\rho + \nabla\rho \cdot v) + \rho[\nabla\partial_t\mu - \partial_t v + D(\nabla\mu)v - (Dv)v]$$
$$+ \rho(\nabla v)(\nabla\mu - v) + \rho(\nabla\mu - v)\text{div}(v)$$
$$= (\nabla\mu - v)(\partial_t\rho + \nabla\rho \cdot v + \rho\text{div}(v))$$
$$+ \rho[\nabla\partial_t\mu - \partial_t v + D(\nabla\mu)v - (Dv)v + (\nabla v)(\nabla\mu - v)].$$

Note that $\partial_t\rho + \nabla\rho \cdot v + \rho\text{div}(v) = \partial_t\rho + \text{div}(\rho v) = 0$, due to the continuity equation. Therefore,

$$\partial_t w + (Dw)v + (\nabla v)w + w\text{div}(v) = \rho[-\partial_t v - (Dv)v - (\nabla v)v + \nabla\partial_t\mu + D(\nabla\mu)v + (\nabla v)(\nabla\mu)]$$
$$= \rho\{-\partial_t v - (Dv)v - (\nabla v)v + \nabla[\partial_t\mu + (D\mu)v]\}.$$

By the stationary condition for the density, $\partial_t\mu + (D\mu)v = 1/2|v|^2$, so $\nabla[\partial_t\mu + (D\mu)v] = (\nabla v)v$, which gives the lemma. $\square$

**Theorem II.1** (Velocity Evolution). *The evolution equation for the velocity arising from the stationarity of the action integral is*

$$\rho[\partial_t v + (Dv)v] = -\nabla U(\phi). \tag{13}$$

*Proof.* This is a combination of Lemmas 1, 4, and 5. $\square$

## D. Stationary Conditions for the Dissipative Case

**Theorem II.2** (Stationary Conditions for the Path of Least Action: Dissipative Case). *The stationary conditions of the path for the action*

$$A = \int [aT(v) - bU(\phi)] \, dt + \int \int_{\mathbb{R}^n} \lambda^T[\partial_t\psi_t + (D\psi)v] \, dx \, dt - \int \int_{\mathbb{R}^n} [\partial_t\mu + \nabla\mu \cdot v] \rho \, dx \, dt, \tag{14}$$

*are*

$$\partial_t\lambda + (D\lambda)v + \lambda\,div\,(v) = b(\nabla\psi)^{-1}\nabla U(\phi) \tag{15}$$
$$a\rho v + (\nabla\psi)\lambda - \rho\nabla\mu = 0 \tag{16}$$
$$\partial_t\mu + \nabla\mu \cdot v = \frac{1}{2}a|v|^2. \tag{17}$$

*Proof.* Note that

$$\nabla[bU](\phi) = b\nabla U(\phi)$$
$$\delta[aT] \cdot \delta\rho = \int_{\phi(\mathbb{R}^n)} \frac{1}{2}a|v|^2\delta\rho \, dx$$
$$\delta[aT] \cdot \delta v = \int_{\phi(\mathbb{R}^n)} a\rho v \cdot \delta v \, dx.$$

Therefore, using (4) and replacing $\nabla U(\phi)$ with $b\nabla U(\phi)$, we have

$$\delta A \cdot \delta\phi = \int\int_{\phi(\mathbb{R}^n)} \left\{ (\nabla\psi) \left[ \partial_t\lambda + \operatorname{div}\left(v\lambda^T\right)^T \right] - b\nabla U(\phi) \right\} \cdot \delta\phi \, dx \, dt,$$

which yields the stationary condition on the mapping. Also, updating (6) yields

$$\delta A \cdot \delta v = \int\int_{\phi(\mathbb{R}^n)} [a\rho v + (\nabla\psi)\lambda - \rho\nabla\mu] \cdot \delta v \, dx \, dt,$$

which yields the stationary condition for the velocity. Finally, updating (8) yields

$$\delta A \cdot \delta\rho = \int\int_{\phi(\mathbb{R}^n)} \frac{1}{2}a|v|^2\delta\rho - (\partial_t\mu + \nabla\mu \cdot v)\delta\rho \, dx \, dt,$$

and that yields the last stationary condition. $\qquad\square$

**Theorem II.3** (Evolution Equations for the Path of Least Action: Dissipative Case). *The evolution equations for the stationary conditions of the action in* (14) *is*

$$\rho[\partial_t(av) + a(Dv)v] = -b\nabla U(\phi). \tag{18}$$

*Proof.* Let $w = \rho(\nabla\mu - av)$ then

$$\partial_t w = (\partial_t\rho)(\nabla\mu - av) + \rho(\nabla\partial_t\mu - \partial_t(av))$$
$$Dw = (\nabla\mu - av)(D\rho) + \rho[D(\nabla\mu) - aDv].$$

Then

$$
\begin{aligned}
\partial_t w + (Dw)v + (\nabla v)w + w\operatorname{div}(v) &= g(\nabla\mu - av)(\partial_t\rho + \nabla\rho \cdot v) + \rho[\nabla\partial_t\mu - \partial_t(av) + D(\nabla\mu)v - a(Dv)v] \\
&\quad + \rho(\nabla v)(\nabla\mu - av) + \rho(\nabla\mu - av)\operatorname{div}(v) \\
&= (\nabla\mu - av)(\partial_t\rho + \nabla\rho \cdot v + \rho\operatorname{div}(v)) \\
&\quad + \rho[\nabla\partial_t\mu - \partial_t(av) + D(\nabla\mu)v - a(Dv)v + (\nabla v)(\nabla\mu - av)] \\
&= \rho\left\{ -\partial_t(av) - a(Dv)v - a(\nabla v)v + \nabla[\partial_t\mu + (D\mu)v] \right\} \\
&= \rho\left\{ -\partial_t(av) - a(Dv)v \right\}.
\end{aligned}
$$

By Lemma 4 and the previous expression, we have our result. $\qquad\square$

### E. Discretization

We present the discretization of the velocity PDE Eqn. 12 (manuscipt) first. In one dimension, the terms involving $v$ are Burger's equation, which is known to produce shocks. We thus use an entropy scheme. Writing the PDE component-wise, we get

$$\partial_t v_1 = -\frac{1}{2}\partial_{x_1}(v_1)^2 - v_2\partial_{x_2}v_1 - \frac{3}{t}v_1 - \frac{1}{\rho}(\nabla U)_1 \tag{19}$$

$$\partial_t v_2 = -\frac{1}{2}\partial_{x_2}(v_2)^2 - v_1\partial_{x_1}v_2 - \frac{3}{t}v_2 - \frac{1}{\rho}(\nabla U)_2, \tag{20}$$

where the subscript indicates the component of the vector. We use forward Euler for the time derivative, and for the first term on the right hand side, we use an entropy scheme for Burger's equation which results in the following discretization:

$$\partial_{x_1}(v_1)^2(x) \approx \max\{v_1(x), 0\}^2 - \min\{v_1(x), 0\}^2 + \min\{v_1(x_1 + \Delta x, x_2), 0\}^2 - \max\{v_1(x_1 + \Delta x, x_2), 0\}^2,$$

where $\Delta x$ is the spatial sampling size, and the $\partial_{x_2}(v_2)^2$ follows similarly. For the second term on the right hand side of (19), we follow the discretization of a transport equation using an up-winding scheme, which yields the following discretization:

$$v_2(x)\partial_{x_2}v_1(x) \approx v_2(x) \cdot \begin{cases} v_1(x_1, x_2) - v_1(x_1, x_2 - \Delta x) & v_2(x) > 0 \\ v_1(x_1, x_2 + \Delta x) - v_1(x_1, x_2) & v_2(x) < 0 \end{cases}.$$

With regards to the gradient of potential, if we use the potential in Eqn. 13 (manuscript), then all the derivatives are discretized using central differences, as the key term is a diffusion. The step size $\Delta t/\Delta x < 1/\max_x\{|v(x)|, |Dv(x)|\}$.

The backward map $\psi$ evolves according to a transport PDE Eqn. 6 (manuscript), and thus an up-winding scheme similar to the transport term in the velocity term is used. For the discretization of the continuity equation, we use a staggered grid (so

that the values of $v$ are defined in between grid points and $\rho$ is defined at the grid points). The discretization is just the sum of the fluxes coming into the point:

$$-\text{div}\,(\rho(x)v(x)) \approx \sum_{i=1}^{2} \left[ -v_i(x) \begin{cases} \rho(x) & v_i(x) > 0 \\ \rho(x + \Delta x_i) & v_i(x) < 0 \end{cases} + v_i(x - \Delta x_i) \begin{cases} \rho(x - \Delta x_i) & v_1(x - \Delta x_i) > 0 \\ \rho(x) & v_1(x - \Delta x_i) < 0 \end{cases} \right],$$

where $\Delta x_i$ denotes the vector of the spatial increment $\Delta x$ in the $i^{\text{th}}$ coordinate direction, $v_1(x)$ denotes the velocity defined at the midpoint between $(x_1, x_2)$ and $(x_1 + \Delta x, x_2)$, and $v_2(x)$ denotes the velocity defined at the midpoint between $(x_1, x_2)$ and $(x_1, x_2 + \Delta x)$. The term $\partial_t \rho(x)$ is discretized with forward Euler. This scheme is guaranteed to preserve mass.