[Reviews · NeurIPS 2018]

Reviewer 1



Summary: The main contribution of this paper is the derivation of an "accelerated" gradient descent scheme for computing the stationary point of a potential function on diffeomorphisms, inspired by the variational formulation of Nesterov's accelerated gradient methods [1]. The authors first derive the continuous time and space analogy of the Bregmann Lagrangian [1] for diffeomorphisms, then apply the discretization to solve image registration problems, empirically showing faster/better convergence than gradient descent. Pros: The paper is well-written. The proposed scheme of solving diffeomorphic registration by discretizing a variational solution similar to [1] is a novel contribution, to the best of my knowledge. The authors also show strong empirical support of the proposed method vs. gradient descent. Cons: 1) Scope of contributions. Although the title and abstract boast "acceleration on manifolds" and "application to diffeomorphisms", it seems to me that this work *only* studies optimization on the *manifold of diffeomorphisms*, rather than general manifold optimization problems. Therefore, the contributions should be compared against similar work in diffeomorphic registration (e.g. image/shape/surface registration). 2) Missing references and baselines. Given 1), some important related work in efficient computation of optimal transport and applications in image registration is missing. In particular, Sinkhorn-type iterations [Cuturi, 2013] are used to compute the proximal operator of the transport problem for large problems, such as indirect image registration in tomography [Karlsson and Ringh, 2017]. The authors should compare their proposed method against this line of work, both in terms of formulation and empirically. While my knowledge of this field is limited, upon some literature search it seems that the Sinkhorn-type methods are very competitive in terms of time complexity and solution quality. Therefore the authors should include a comparison with such alternative methods. 3) Computational complexity and theoretic analysis. While the proposed method is named "accelerated gradient descent", it is not shown to provably accelerate gradient descent or even converge to a stationary point by the authors' discretization scheme. The authors only propose a discretization method (line 255 and supplementary E) but omit the analysis of its convergence and iteration complexity. To sum, this work contains some novelty but lacks rigorous analysis of its proposed method; more importantly, it misses theoretical and empirical comparison to some important related work of the same application, making it impossible to judge the significance of the proposed method at this point. [Cuturi, 2013] Cuturi, M., 2013. Sinkhorn distances: Lightspeed computation of optimal transport. In Advances in neural information processing systems (pp. 2292-2300). [Karlsson and Ringh, 2017] Karlsson, J. and Ringh, A., 2017. Generalized Sinkhorn iterations for regularizing inverse problems using optimal mass transport. SIAM Journal on Imaging Sciences, 10(4), pp.1935-1962.

Reviewer 2



Inspired by recent work that have shown that the continuum limit of inertial schemes for gradient descent may be formulated with variational principles, this paper extends the idea to general manifolds, with an emphasis on the infinite dimensional manifold of diffeomrophisms. The main contributions are: 1) a variational approach to inertial smooth optimization on manifolds; 2) a Riemannian metric tailored to produce inertial schemes on diffeomorphisms, which is the kinetic energy of a distribution of moving mass; 3) A numerical discretization scheme which requires entropy schemes. The approach is illustrated on several computer vision and image processing tasks including registration and optical flow. The contribution is very interesting and original enough. Having said, I have several comments: 1) The authors keep talking from the very beginning about accelerated schemes, but this terminology quite inadequate here (in fact unjustified). First, in the Euclidean case, such terminology is legitimate according to a complexity (lower) bound argument on the class of convex smooth functions (with Lipschitz gradient). Here, given the setting on general manifolds, there is no such complexity bound. Moreover, and more importantly, the authors did not provide any convergence rate analysis on any appropriate quantity (e.g. poytential energy), neither in the continuous nor in the discrete settings. This is mandatory to be able to state formally that the presented scheme leads to acceleration (with dissipation, which is known to lead to inertia/accleration in the Hilbertian case). I would then suggest to completely rephrase this. 2) In relation to 1), in the experiments, it would be wise to report the convergence profiles in log-log scale of the potential energy to be able to read directly the slopes. 3) I think that one of the most important parts to the NIPS audience would be the discretization, which is postponed to the supplementary material. 4) In relation to 1) and 3), if one is able to prove a convergence rate (on e.g. the potential) in the continuous case, it would be much more convincing to see a result stating that the proposed discretization preserves that rate too. 5) There are a few typos such as P6, L227: epsilon lacking in the limit. 5) Section 4: regularity parameter alpha -> regularization parameter alpha. 7) Figure 3: in the Hilbertian case, it is known that the upper-bound on the descent step-size is twice larger for GD than AGD. It would be interesting to discuss this aspect in the Riemannian setting here.

Reviewer 3



1. A tautology in line 96 "The model space may be finite or infinite dimensional when the model spaces are finite or infinite dimensional, respectively." 2. The wrong definition of Riemannian manifold, it should be smoothly varying inner product, not just any metric. Furthermore, if f_p is not assumed to be smooth (at least differentiable), one can't define metric! Rewrite the definitions. In general, I understood the overall idea presented in this paper, I verified the proof of Thm. 3.1, but not sure about the other proofs, though the statement of the theorems looks correct. A nice paper overall. ------------------------------------------------------------------------------------------ POST REBUTTAL COMMENTS: I am overall satisfied with this work and the rebuttal, hence I vote for an acceptance of this paper.

Reviewer 4



In this paper, the authors consider acceleration schemes when optimising functionals over Diff(R^n), the space of diffeomorphisms on R^n. The method is an analog to more classical approaches for of accelerated optimisation schemes on finite dimensional manifolds. The authors motivate the need to optimise functionals on Diff(R^n) by quoting applications to optical flows. The experiment section does a good job to study the effect of various hyper-parameters and set-up choices. Most experiments are toy experiments which make it easy to compare results with what one would expect from them. On the other hand, the only non-toy experiment is hard to analyse (it's really hard to tell if the found diffeomorphism is good, or even if it is better than the one found from normal gradient descent). I enjoyed reading the paper, and found it handled well the treatment of very complicated mathematical objects. I nevertheless think that the paper would benefit from some more introductory references to some of these complicated subjects (see comments below). Comments: 1. line 45, I don't get point 3. The kinetic energy is a functional, not a metric. Do the authors mean that this kinetic energy corresponds to a norm, and a Riemannian metric can be derived from this norm? But in that case, this is a family of metrics, since the kinetic energy (4) is time dependent: it depends on \rho, which is time dependent (see line 182). 2. line 124: don't you also need \gamma(0, s) = constant, and \gamma(1, s) = constant, in order to keep endpoints fixed. 3. line 163: this needs a reference here. There are many flavours of infinite dimensional manifolds (e.g.: Banach, Frechet). Also, with this definition of the tangent space, the kinetic energy (4) is not defined (in the sense of being finite) for all tangent vectors: just make $|v|$ go to infinity fast enough compared to $\rho$. 4. line 178: \rho should take values in R_+ 5. after line 183, formula (3): it would be clearer if \rho was replaced by rho_t everywhere in this formula, to make the time dependence clearer 6. after line 187, formula (4): it would be clearer to write $T(v, \rho)$ instead of $T(v)$ since this depends on \rho. Also, the integral subscript $\varphi(R^n)$ could be simplified to $R^n$ since $\varphi$ is a diffeomorphism of $R^n$. 7. after line 201, formula (6): I understand why this is equivalent to (2), but why is it better to use the inverse diffeomorphisms? 8. line 221: the sentence is mal-formed. 9. line 227: should $v$ be replaced by $v_\epsilon$ inside the limit? 10. line 264: the term 'reasonable' sized image is subjective. I'd suggest saying 'larger sized images' instead. 11. All experiments use a displacement that is smaller than the size of the squares. Is that because otherwise there is no gradient (as in, equals 0) if this condition is not satisfied. This would be good to clarify this. 12. All experiments use different values for $\alpha$, ranging from 0.02 to 8. Why change it so much, and how were these values chosen? 13. It would be good to have some visualisations of what diffeomorphisms was found by the method (for ex, for square translation, was this just a translation?). One possibility would be to create an image with some colour gradient along both the x and y directions, and show how the images look like after applying the diffeomorphisms. This would also help understand what sub-optimal solution was found by normal gradient descent for the 'square translation + non-uniform scaling' experiment. 14. The 2 MR cardiac images are extremely close to each other to start with. I personally see no qualitative difference between the solutions found by gd and agd. I also am not able to judge if any of these solutions is good. Having experiments on real data is good, but I don't find that this one is particularly useful. 15. In the Appendix, section E. Discretization, formula (20): $v_1\partial_{x_1}v_1$ should be $v_1\partial_{x_1}v_2$ ============ Thank you to the authors for their reply. After reading the other reviewers opinion, I will maintain my score and positive opinion of the paper, but I would like to emphasise a very valid point made by other reviewers: the term 'accelerated' is misleading. The algorithm proposed by the authors is inspired by 'accelerated' methods, but without theoretical bounds, I think this term should be used with more caution. In particular, I believe it should probably be dropped from the title. It can still be used in the paper when providing empirical evidence of acceleration, or when referring to other accelerated methods.